# Behavioural Determinants of Appropriate Antibiotic Prescribing for Urinary Tract Infections in Nursing Homes: A Qualitative Study of Stakeholders’ Perspectives

**DOI:** 10.3390/antibiotics15010005

**Published:** 2025-12-19

**Authors:** Indira Coenen, Sien Lenie, Kristien Coteur, Carmel Hughes, Veerle Foulon

**Affiliations:** 1Department of Pharmaceutical and Pharmacological Sciences, KU Leuven, 3000 Leuven, Belgium; 2Department of Public Health and Primary Care, KU Leuven, 3000 Leuven, Belgium; 3School of Pharmacy, Queen’s University Belfast, Belfast BT7 1NN, Northern Ireland, UK

**Keywords:** antimicrobial resistance, antimicrobial prescribing, antimicrobial stewardship, nursing homes, urinary tract infections, behaviour change research

## Abstract

**Background/Objectives**: Urinary tract infections (UTIs) are the primary indication for antibiotic use in nursing homes (NHs); yet inappropriate prescribing, including incorrect initiation, excessive prophylactic prescribing and prolonged treatment duration, is common. This study aimed to identify key determinants of appropriate antibiotic prescribing for UTIs in NHs by exploring the behaviours and perspectives of relevant stakeholders. **Methods**: Interviews and focus group sessions with regard to a purposive sample of 4 NHs and healthcare professionals were conducted between June 2023 and April 2024 in Flanders (Belgium). The topic guide was developed based on the Theoretical Domains Framework (TDF). A combination of deductive and inductive coding was used to identify behavioural determinants within each TDF domain. Key behavioural determinants were identified based on their importance, relevance, and feasibility. **Results**: We conducted 31 semi-structured interviews with residents/relatives (*n* = 13), physicians (*n* = 9), pharmacists (*n* = 10), and NH management (*n* = 5) and held 4 focus group sessions with nurses (*n* = 16) and nurse aides (*n* = 10). Appropriate antibiotic prescribing for UTIs in NHs was influenced by a complex interplay of behavioural determinants. Key behavioural determinants included lack of knowledge of guidelines, lack of self-reflection and monitoring, fear of missing complications, feelings of powerlessness, prioritising residents’ comfort, hierarchical relations with treating physicians being dominant, social pressure to prescribe, and the NH as a challenging context. **Conclusions**: This study identified key behavioural determinants that should be targeted to optimise antibiotic prescribing for UTIs in NHs. These findings underscore the need to conduct a theory-informed, multifaceted intervention to support behaviour change across professional roles and improve antimicrobial stewardship in this setting.

## 1. Introduction

Antimicrobial resistance (AMR) is one of the most urgent threats to global health [1]. Although antimicrobials have saved millions of lives, their widespread and often inappropriate use has led to the emergence of multidrug-resistant microorganisms, causing severe infections with uncertain outcomes and high treatment costs [2]. Without targeted action, it has been projected that AMR will cause 10 million deaths annually by 2050 [2]. In response, different (inter)national authorities, including the World Health Organization, have developed action plans that call for urgent and coordinated measures [3,4]. Antimicrobial stewardship (AMS) programs—which support the optimal choice, dosage, route, and duration of antimicrobial therapy—are a key component of these action plans [5].

Nursing homes (NHs) represent a particularly critical setting for AMS. Residents are frail, often with multiple comorbidities, and highly susceptible to infections [6]. Based on a recent European point-prevalence study [7], the crude point prevalence of residents receiving at least one antimicrobial was 4.1%. Treatment was the most common indication for antimicrobial prescribing (68.8%), followed by prophylaxis (29.1%). Prophylaxis was primarily prescribed for urinary tract infections (UTIs) (68.5%), while treatment was most frequently initiated for UTIs (41.8%), respiratory tract infections (30.5%), and skin and skin-structure infections (15.4%). In Belgium, more than half of all antimicrobial prescriptions were related to urinary tract infections (UTIs) [8,9], and inappropriate use was highly prevalent, with incorrect treatment initiation, excessive uroprophylaxis, and unnecessarily prolonged treatment durations being the most common deviations from the national guidelines [9].

Prescribing decisions for UTIs in older adults are shaped by a sequence of interrelated steps involving multiple stakeholders [10]. First, a UTI is suspected by residents, relatives, or nursing staff. Second, the physician considers the diagnostic work-up, often constrained by difficulties in obtaining urine samples, logistical and financial barriers, and limited access to timely laboratory testing. Finally, the decision to prescribe antibiotics is taken, frequently influenced by the expectations of relatives or staff [10,11]. This complex interplay of medical, organizational, and social factors increases the risk of inappropriate antibiotic use [10].

Evidence-based guidelines for management of UTIs exist in Belgium, yet their implementation in NHs remains limited [9]. A previous study in Belgium has explored the views of coordinating physicians (CPs)—general practitioners designated to coordinate medical activity in NHs—on AMS interventions in NHs. Education was considered the cornerstone for any future development. The top priority identified was to reduce unnecessary treatment of asymptomatic UTIs [12]. However, as antibiotic prescribing is preceded and shaped by the actions of diverse stakeholders, a broader multi-perspective understanding of behavioural determinants is essential. The Theoretical Domains Framework (TDF) provides a structured, theory-driven approach to investigating such determinants, condensing 112 behavioural constructs into 14 domains that map onto the Capability, Opportunity, Motivation—Behaviour (COM-B) model [13]. Applying these frameworks in the early stages of intervention development, as recommended by the Medical Research Council’s Complex Intervention Framework, helps ensure that strategies address the most relevant barriers and enablers to change [14].

This study therefore investigates behavioural determinants influencing appropriate antibiotic prescribing for UTIs in NHs in Belgium, considering the perspectives of a range of stakeholders. By systematically mapping these determinants using the TDF, it aims to generate insights for the design and evaluation of interventions optimising antibiotic prescribing in NHs.

## 2. Results

### 2.1. General Characteristics and Antibiotic Policy of NHs

Thirteen NHs were approached, resulting in four participating NHs from three provinces in Flanders, each affiliated with one of three predefined hospital outbreak support team networks. Nine NHs declined participation due to time constraints. The questionnaire on NH characteristics and antibiotic policy was completed by the NH management, with some input from nurses.

One NH was a public facility in an urban area, while the other three were private and located in rural areas. Two were large (>180 beds), and two were medium-sized (90–180 beds). Over 80% of residents in all NHs had high care dependency (Katz-scale category B, C or D). All NHs had a CP and one NH also employed a coordinating pharmacist. Two NHs were supplied with medication by a local and independent pharmacy, and two were supplied by a pharmacy that was part of a network of pharmacies (Table 1). 

Regarding antibiotic policy, one NH reported having its own therapeutic guidelines for appropriate prescribing in respiratory and skin and skin-structure infections, whereas none reported having their own specific guidelines for UTIs. Two NHs used a therapeutic formulary to guide prescribing practices. In two NHs, suspected infections were initially monitored by observing clinical signs and symptoms, including temperature. In the other two NHs, initial steps involved diagnostic tests such as urine dipstick tests or microbiological testing, with both approaches routinely used in three NHs. In cases of resistant or transmissible pathogens (e.g., *Clostridioides difficile*), treating physicians (TPs) were reported to consistently inform the CP or NH management, although one NH noted that this was done “most of the time.”

### 2.2. Participants’ Characteristics

A total of 31 interviews and 4 focus groups were conducted, involving 63 participants including pharmacists, TPs, CPs, nurses, nurse aides, quality coordinators, NH directors, residents, and relatives. Interviews with healthcare professionals (HCPs) had a mean duration of 60 min, focus groups 90 min, and interviews with residents or relatives 30 min. Participant demographics are summarised in Table 2. Participants were predominantly female (except physicians), with work experience ranging from <1 to >20 years across most stakeholder categories, while TPs all had ≤3 years and NH managers ≥ 14 years of experience.

### 2.3. Overall Perspective on Appropriate Antibiotic Prescribing for UTI in NHs

Most participants reported inappropriate antibiotic prescribing for UTI in their NHs, mainly linked to prescribing practices of visiting TPs, rather than the antibiotic policy in the NHs themselves. Deviation from guidelines was common, with inappropriate prescribing characterised by incorrect indication, choice, and prolonged duration of the antibiotic treatment.

The interviewed pharmacists particularly noted frequent prescriptions for fosfomycin, nitrofurantoin, and quinolones. Chronic antibiotic prescribing (>three months treatment duration) for the prevention of UTIs was also observed, with some pharmacists reporting that stop dates were occasionally overlooked.

### 2.4. Determinants of Appropriate Antibiotic Prescribing for UTI in NHs

The data showed that appropriate antibiotic prescribing, although recognised as the responsibility of physicians, was influenced by a complex interplay of behavioural determinants, with the nursing team—including nurses and nurse aides—playing a central role. Appendix A in the Appendix A outlines the coding tree, presenting each theme (TDF domain) along with the subthemes (behavioural determinants) that emerged from the analysis.

Upon analysis, we identified key behavioural determinants that should be targeted to optimise antibiotic prescribing for UTIs in NH. These key determinants are organised according to the TDF domains and are illustrated with representative quotes in Table 3.

We summarised the identified key behavioural determinants for appropriate prescribing for UTI in NHs in Figure 1. These determinants were categorised across three levels [the micro level (individual stakeholders), the meso level (interpersonal dynamics), and the macro level (the broader context of the NH setting in Flanders)] and have been described below using these levels.

#### 2.4.1. Key Determinants at the Micro-Level

For each stakeholder, we describe how their role was currently undertaken and which behavioural determinants appeared to influence this at the personal or micro level. These key behavioural determinants correspond to the TDF domains Emotion, Goals, Knowledge and Memory attention and decision process.

##### The Nursing Team—Central Role but Need for Empowerment

The interviews with all participants revealed that the nursing team played a central role in the prevention, diagnosis, and follow-up of UTIs in NHs. Nurse aides, responsible for daily intimate care, were described by nurses as their “eyes and ears.” Their observations were also highly valued by other HCPs and NH managers. A nurse highlighted the critical role of nursing staff in facilitating open communication about UTIs. She noted that a degree of stigma still surrounded the condition and its potential causes, which may discourage residents from consistently reporting symptoms.

The data showed that nurses functioned as gatekeepers of information and key communicators among stakeholders. Despite this essential role, many reported feeling powerless. A major barrier was the lack of knowledge regarding current evidence for UTI management. The absence of clear, local protocols further undermined their confidence and limited their capacity for critical thinking and dialogue with physicians. This was reflected in the continued use of dipstick tests and the focus on non-specific symptoms such as urine odour and confusion. Interviewed nursing teams emphasised that maintaining residents’ comfort was their foremost priority.

“*Nurses also play a very important role, because they are usually the first caregivers in the NH to detect or suspect a UTI.*”(NH3TP)

##### Residents and Relatives—Prioritising Comfort

Interviews with residents and relatives showed that UTIs placed a significant burden on residents. A resident, who suffered from recurrent UTIs, expressed feeling desperate and not understanding the cause of the recurring infections. Residents perceived antibiotics as essential for pain relief and expected prompt action from HCPs. Likewise, relatives prioritised rapid symptom resolution. The interviewed residents indicated that they were well-monitored by the nurses and that they did not always expect a physician to be present at the initiation of antibiotic treatment. The experiences of relatives of residents with dementia varied. Some expressed disappointment with the HCPs when they only learned from the expense report—which lists all costs, including the medication administered—that a UTI had occurred. Others, however, voiced complete trust in the HCPs and did not expect to be contacted at every suspicion of a UTI. The perceived short-term benefits of antibiotic prescribing—comfort and prevention of complications—outweighed long-term concerns, such as AMR, particularly among relatives, who expressed the most fear about potential deterioration.

“*As long as it helps me. That’s the main thing*.”(NH1Res2)

##### Treating Physician—Ultimate Responsibility but Difficult to Access

TPs were described as playing a key role in diagnosis, prescribing, and treatment follow-up, yet their limited presence and availability in NHs often hindered involvement.

“*If, as a GP, you have only one patient in that NH and you would have to go there specifically for that one patient for something like this [a UTI], then the temptation is great to try to handle quite a lot over the phone.*”(CP1)

Interviewed TPs reported a lack of self-monitoring and critical reflection, which contributed to more habitual prescribing behaviours. While aware of the population’s vulnerability, TPs acknowledged that fear of complications often led them to prescribe antibiotics more readily than following guidelines.

##### Pharmacist—Potential Role in Auditing and Feedback

Pharmacists, while primarily involved in dispensing of antibiotics, were seen as having potential to contribute to audit and feedback. Interviewed pharmacists reported sporadic involvement in retrospective feedback at the resident level (medication reviews for chronic medications) or institutional level (benchmarking). Most pharmacists recognized the value of expanding this advisory role. Pharmacists also noted their role in providing education and practical medication-related information. According to most participating HCPs, the current role of the pharmacist in UTI care was largely limited to the dispensing of antibiotics, without critical reflection on whether the prescribed therapy was optimal for the resident. This limited role was primarily due to several macro-level barriers that prevented pharmacists from taking on a more active role in AMS (see 2.4.3). One pharmacist cited prospective feedback at the point of prescribing as the ideal, although currently unfeasible.

“*Well yes, my role today during those multidisciplinary consultations is to provide some training about the problem and also to explain the guidelines, especially at the moments when we often see them [deviations from the guidelines] in the [medication] schedules. For clarity, we are not going to do that for every patient. And later, to intervene more directly with the physician when they prescribe something and discuss it. That is the ultimate role: to be able to do that as a team.*”(P2)

##### Coordinating Physician and Nursing Home Management—Potential Role in Supporting Collaboration

NH management and CPs were identified as key facilitators of interprofessional collaboration. Managers emphasized residents’ comfort as a core goal but admitted lacking insight into overall antibiotic prescribing—resulting in low prioritization. CPs similarly lacked awareness of their colleagues’ prescribing practices, limiting their coordination role in AMS efforts. According to the interviewed physicians and pharmacists, CPs had a (potential) role in policy development, possibly together with pharmacists. Coordinating physicians and pharmacists frequently referred to the composition of the emergency medication kit as an influence on antibiotic policy; this is a small stock of essential drugs, often including antibiotics, that enables immediate initiation of treatment until a prescription is obtained or the pharmacy delivers the medication.

“*I cannot interfere in the treatment of other doctors, you know. So I only come here and sometimes I can take a look, but when you are a CP in a NH, you are often advised to develop general guidelines and general strategies.*”(NH3CP)

#### 2.4.2. Key Determinants at the Meso-Level

At the meso level, social dynamics between stakeholders emerged as a key influence on appropriate antibiotic prescribing for UTIs, reflecting the TDF domain Social influence.

##### Social Pressure to Prescribe Antibiotics

Interviewed HCPs identified social pressure to prescribe antibiotics as a key barrier to appropriate prescribing. Physicians specifically reported feeling pressured by residents, relatives, and nursing staff, while nurses experienced pressure primarily from relatives. Relatives were described as highly vigilant, often prompting HCPs to act more quickly. In cases involving residents with cognitive impairment who were unable to articulate symptoms, nurse aides or relatives frequently reported signs such as foul-smelling urine or confusion—symptoms not supported by current evidence as reliable indicators of UTIs. These observations were passed on to nurses, who functioned as the central communication link between staff, residents, relatives, and other HCPs, and consequently experienced additional pressure in this intermediary role.

“*People [residents] can’t… There are maybe, very occasionally, some who can still indicate it [the symptoms of UTI]. But there are very few who can still express it themselves.*”(NH2N3)

##### Hierarchical Relations with Treating Physicians on Top

Although participating TPs were influenced by their social environment (nurses, residents and relatives), other HCPs reported being strongly influenced by physicians. As mentioned above, TPs were seen as holding ultimate responsibility for clinical decisions and were perceived as having the authority to deviate from guidelines at their discretion. Residents and relatives also indicated that their beliefs about the consequences of antibiotic prescribing were shaped by physicians’ opinions. This hierarchical dynamic was reflected in the pronounced deference exhibited by other HCPs (non-physicians). Moreover, CPs reported that they experienced providing feedback to fellow physicians as delicate, which positioned TPss in a relatively isolated role: they were neither questioned by other HCPs nor by CPs. Both nurses and pharmacists emphasized that physicians often remain unreceptive to input, reinforcing their dominant position in the decision-making process.

“*But that is mainly the older generation [of physicians] who are not open to [feedback], and the younger ones are. That’s the difference.*”(NH3N1)

“*Yes, if the physician decides that antibiotics have to be started, then we usually just go along with that.*”(NH2P1)

#### 2.4.3. Key Determinants at the Macro-Level

At the macro level, several contextual factors related to the broader circumstances in which participants worked and lived influenced the appropriateness of antibiotic prescribing for UTIs, aligning with the TDF domain Environmental context and resources.

##### Nursing Home as a Challenging Context

Implementing an appropriate antibiotic policy for the prevention, diagnosis, and treatment of UTIs in NHs was perceived as challenging due to multiple inherent barriers. The frailty of residents, combined with diagnostic uncertainty, created anxiety among HCPs and relatives about potentially missing infections or failing to act promptly. Staffing shortages—particularly among nurses—were reported as a major constraint. The involvement of numerous TPs per NH, together with the limited mandate of CPs, was seen as an obstacle to establishing a uniform policy, weakening interprofessional collaboration, reducing accessibility, and often resulting in TPs being physically absent from the NH.

HCPs further highlighted the absence of an integrated interprofessional communication system, the lack of structural (including financial) support for interprofessional consultations in which a local antibiotic policy could be developed and operationalized, the absence of antibiotic prescribing monitoring, and the lack of a clear, accessible guideline as important macro-level barriers to appropriate prescribing.

“*If they would also support the NHs a bit more for those electronic things—general practitioners get a budget for that; we get nothing.*”(NH1M)

“*But ideally you would simply have a physician in the NH who can quickly go and have a look and who applies consistent policy, but that’s not how our NHs are structured, you know?*”(NH3CP)

Macro-level factors also limited the pharmacist’s role primarily to dispensing (cf. earlier description of the pharmacist’s role at the micro level). Participating pharmacists described several barriers to greater involvement in medication monitoring, including limited knowledge of the current clinical situation and medical history of residents, the absence of a unified guideline, and insufficient time and funding. These constraints undermined pharmacists’ confidence and reduced opportunities for meaningful participation in AMS initiatives.

## 3. Discussion

In this qualitative study, we explored behavioural determinants of antibiotic prescribing for UTIs in NHs using the TDF, including the perspectives of HCPs, NH management, residents, and relatives. Participants consistently expressed intentions to follow guidelines and collaborate interprofessionally. However, despite these intentions, consistent implementation of guidelines and interprofessional collaboration remained limited. Residents’ expectations of rapid relief, the painful nature of UTIs, and HCPs’ fear of missing complications often acted as drivers of unnecessary prescribing.

Our findings indicate that antibiotic prescribing in NHs was shaped by interacting micro-, meso-, and macro-level determinants. At the micro level, key behavioural determinants included lack of knowledge on guidelines (Knowledge), lack of self-reflection and monitoring (Memory, attention and decision process), fear of missing complications (Emotion), feelings of powerlessness (Emotion) and prioritising residents’ comfort (Goals). At the meso level, we identified two key behavioural determinants: hierarchical relations with TPs being dominant (Social influence) and social pressure to prescribe (Social influence). At the macro level, numerous identified barriers reflected a structurally challenging NH setting (Environmental context and resources).

These results align with previous qualitative research showing the strong influence of NH contextual and social factors on prescribing decisions in NHs [10,15,16]. A recently published Australian study identified determinants to the implementation of infection prevention and control practices in NHs using semi-structured interviews with HCPs [17]. Similar to our approach, the authors applied the TDF to guide data collection and analysis. Their findings likewise identified determinants across multiple levels: at the micro-level, factors such as knowledge about the importance of infection prevention and control played a role; at the meso-level, social influences—including competing priorities—emerged as relevant; and at the macro-level, access to necessary resources, such as hand hygiene products, was highlighted as a critical enabling factor. Consistent with earlier studies in the NH setting [10,16], nursing staff, residents and relatives can influence antibiotic prescribing decisions. In our study nurses emerged as central intermediaries between residents, families, and physicians, yet their role was often constrained by workload, unclear responsibilities, and limited communication structures. Pharmacists, described as important AMS actors in NHs elsewhere [18,19], had a less visible role in Belgium, largely restricted to supplying antibiotics, underscoring the need for context-sensitive strategies. In a recent published study of Lambert et al. [20], the high—yet underused—potential of community pharmacists to optimise antibiotic use was highlighted.

By incorporating residents’ and relatives’ perspectives—rarely included in earlier work [11,21] our study revealed that UTIs had substantial impact on well-being and quality of life, with expectations for immediate action. Ahouah et al. [22] examined perceptions of antibiotic therapy among NH residents and nursing staff, reporting misconceptions among residents regarding AMR and a predominant reliance on physicians for antibiotic-related information; these findings broadly align with our study. However, whereas Ahouah et al. found that nurses were not regarded as credible sources of antibiotic information, our data suggest that residents had considerable trust in nursing staff and did not consistently perceive physician involvement as necessary for UTI management [22]. While relatives sometimes exerted pressure to prescribe, more often they felt insufficiently involved in decision-making. These insights highlight the value of communication strategies such as safety-netting. In practice, safety-netting entails clear guidance on vigilant monitoring of the residents’ parameters and alarming symptoms to ensure residents’ safety, which can reduce anxiety and align expectations across stakeholders.

An additional contribution of our study is the explicit use of the TDF to identify psychological and affective mechanisms underlying prescribing behaviour. This responds to the call from Singh et al. [23] for more attention to understanding how HCPs, especially NH staff, feel about implementing AMS strategies and other peoples’ (e.g., residents and relatives) perceptions of stewardship. Fear, uncertainty, and powerlessness were prominent in our data, suggesting that behaviour change strategies should go beyond knowledge provision and explicitly address emotions and risk perceptions.

Strengths of this study include its multi-stakeholder design and the inclusion of diverse perspectives from all stakeholders directly involved in antibiotic prescribing in NHs. Although all data were collected through interviews or focus groups, variation in stakeholder groups allowed us to construct a more accurate and nuanced understanding of behaviours and their determinants. Rigorous application of the TDF framework in the design, data collection, and analysis, combined with independent coding by two researchers, further enhanced the credibility of findings. The inclusion of residents and relatives, including those of people with dementia, added perspectives often absent from previous research.

Limitations should also be noted. Only four NHs were included, although they were carefully selected, which limited the overall number of interviews with residents and relatives. Relatives, in particular, expressed distinct perspectives regarding their expectations of HCPs and their own involvement, suggesting that data sufficiency for this group may not have been fully achieved. Social desirability bias and Hawthorne effects may also have influenced responses; however, participants generally appeared to speak openly once trust was established. The fact that the interviewer IC is a pharmacist may have encouraged socially desirable responses regarding the importance of appropriate antibiotic use, as many HCPs emphasized positive intentions to collaborate and support AMS. Interviews with residents yielded only limited depth due to awareness and communication barriers. Finally, some findings may be context-specific: while many insights are transferable, results related to specific professional roles should be interpreted with caution outside the Flemish context.

Taken together, these findings underscore the need for AMS interventions in NHs that are targeted to the different stakeholders, multifaceted, and tailored to the specific context. Interventions should strengthen psychological capability (knowledge, reflective decision-making), improve social and physical opportunities (clear role delineation, communication systems, interprofessional collaboration), and address motivation (reducing fear, balancing comfort and safety). Policy-level support, including clear national guidelines, integrated communication structures, and formalised roles for CP and pharmacists, will be crucial to sustain behaviour change.

## 4. Methods

### 4.1. Study Design

We used a qualitative approach, including interviews and focus groups, to gain detailed understanding of HCPs (physicians, pharmacists, nurses and nurse aides), NH management, and residents’ and relatives’ perspectives on the appropriate prescribing of antibiotics in NHs in Flanders (Belgium). The topic guides (see Appendix A) were developed based on the TDF and refined after pilot testing.

All key components of this qualitative study have been reported in accordance with the COREQ checklist (see Appendix A). KC, VF and CH—senior researchers with extensive experience in qualitative research—collaborated closely with junior researchers IC and SL. All researchers are female. IC, VF, and CH are pharmacists by training, with a specific interest in the appropriate use of medications in NHs.

### 4.2. Participants

NHs were selected using purposive sampling to ensure variation in the number of beds, organisational type (private or public), geographic location, and affiliation with hospital outbreak support team networks. Additionally, the sampling strategy aimed to include at least one NH supplied by a local pharmacist and one located in a city centre. Participants were recruited until data sufficiency was reached.

The study invitation was initially sent by mail to four NHs that met these criteria to ensure a balanced sample. If a NH declined participation, another NH with similar characteristics was contacted. This process continued until four NHs were successfully recruited.

To gain a comprehensive understanding of the target behaviour from a multi-stakeholder perspective, semi-structured interviews were conducted within each NH with the CP, a visiting TP, the pharmacist who supplied medication (i.e., the supplying pharmacist), and a member of the NH management (either the director or the quality coordinator). Additionally, residents who had recently used antibiotics to treat (recurrent) UTIs were interviewed. For residents with dementia, relatives were also invited to participate in the interview. Per NH, one focus group with nurses and nurse aides was organised.

Coordinating pharmacists—who are responsible for developing and monitoring the NHs’ medication policy, with particular focus on the rational use of medicines, including antibiotics—and additional CPs, who were not affiliated to the participating NHs, were recruited from the researchers’ professional network.

### 4.3. Data Collection

Data collection began with a member of NH management completing an online questionnaire via Qualtrics, which included questions on general characteristics and the antimicrobial policy of the NH. Subsequently, the managers were asked to assist in recruiting participants for the study.

Interviews and focus groups were conducted between June 2023 and April 2024 by the researchers (IC, SL, and VF), with Master’s students in Pharmaceutical Care as observers, who, at the end of an interview or focus group, shared their main observations with the participants to allow for participant feedback. Audio recordings for in-person interviews in the NHs were made using a smartphone’s audio recorder, while recordings for online interviews were captured via Microsoft Teams. Following each session, recordings were removed from the smartphone and securely stored within KU Leuven’s Microsoft Teams environment, exclusively for transcription purposes.

All interviews were transcribed verbatim, and the audio recordings were permanently deleted after transcription. To ensure participant confidentiality, the transcripts were pseudonymised (using a code that indicated the participant’s affiliated nursing home and stakeholder category) during the transcription process. Field notes, that were written immediately after each interview or focus group, were placed at the beginning of the transcripts to provide context for the conversation and to summarise the key points that emerged. All interviews were conducted in Dutch. For reporting purposes, quotes were translated into English. To ensure the accuracy and preservation of the original meaning, all translations were reviewed and validated by a native English speaker proficient in Dutch.

### 4.4. Data Analysis

Descriptive analyses were conducted for the NH questionnaire data. The interviews and focus groups were analysed using an iterative approach, supported by NVivo 14 software.

In the first phase, aimed at identifying behavioural determinants, the researchers conducted an in-depth familiarisation with the data by thoroughly reading and re-reading and summarising the transcripts. Subsequently, two researchers (IC, SL) independently coded the data using the 14 domains of the TDF, following a deductive framework analysis approach. After coding seven transcripts and intermittent discussions, the researchers achieved a shared understanding of the TDF domains—supported by an acceptable inter-rater reliability score (Cohen’s kappa = 0.49). The remaining transcripts were divided between the two researchers for coding by one researcher (IC or SL). Regular team meetings were held to resolve any coding uncertainties, facilitated by a third researcher experienced in qualitative analysis (KC). Emerging themes within each TDF domain, representing behavioural determinants, were then identified through inductive content analysis by IC, SL and KC.

In the second phase, focused on identifying key behavioural determinants, the research team reviewed the behavioural determinants within each TDF domain. This was followed by discussions within the research team to evaluate the importance and relevance of the identified behavioural determinant in relation to the target behaviour. Additionally, the feasibility of overcoming the identified barriers or enhancing facilitators within the NH setting was assessed. The final step involved a consensus-meeting to determine the key behavioural determinants for a future intervention. These key determinants were categorised across three levels [the micro level (individual stakeholders), the meso level (interpersonal dynamics), and the macro level (the broader context of the NH setting in Flanders)].

## 5. Conclusions

In conclusion, this study extends existing evidence by integrating multiple stakeholder perspectives and applying a behavioural framework to antibiotic prescribing for UTIs in NHs. The results highlight that appropriate prescribing is not only a matter of knowledge but also of emotions, social dynamics, and systemic constraints. Future interventions should therefore move beyond education alone and support complex behaviour change within an enabling environment, ultimately aiming to optimise antibiotic prescribing and improve care quality for the vulnerable NH population.

## Figures and Tables

**Figure 1 antibiotics-15-00005-f001:**
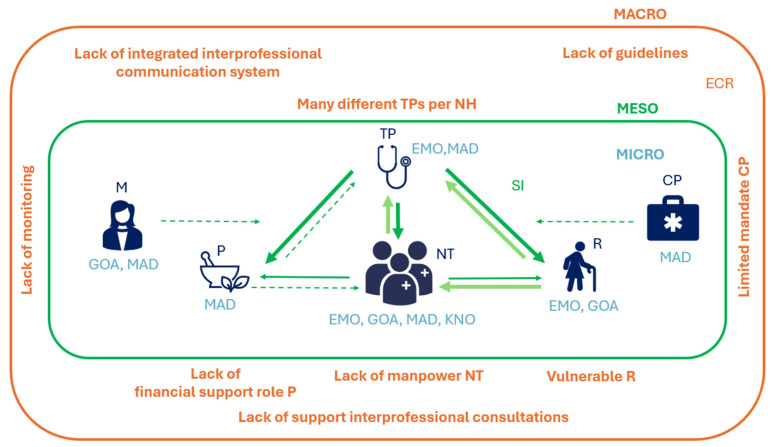
Key behavioural determinants for appropriate prescribing of antibiotics related to UTIs in NHs. TDF domains at the micro-level (blue): EMO = Emotion, KNO = Knowledge, GOA = Goals, MAD = Memory attention and decision processes; TDF domain at the meso-level (green): SI = Social influence, with light green arrows indicating social pressure to prescribe, dark green arrows representing hierarchical relations, and dashed arrows indicating a lack of social influence; TDF domain at the macro-level (orange): ECR = Environmental context and resources. Stakeholders: NT = nursing team, P = pharmacist, TP = treating physician, R = residents and relatives, CP = coordinating physician, M = nursing home management.

**Table 1 antibiotics-15-00005-t001:** Nursing homes’ general characteristics.

Characteristics	Nursing Home 1	Nursing Home 2	Nursing Home 3	Nursing Home 4
Location	Rural	Rural	Urban	Rural
Organisation status	Private, non-profit	Private, non-profit	Public	Private, for profit
Number of beds	>180	90–180	>180	90–180
Percentage of residents categorised as having high care dependency (%)	86	93	82	90
FTE nurses	ND	21	25	17
FTE nurse aides	ND	40	49	27
CP	Yes	Yes	Yes	Yes
Number of visiting TPs	40	40	ND	12
Supplying pharmacy	Local independent pharmacy	Independent pharmacy, specialised in delivering medications to NHs	Supplying pharmacy is part of a pharmacy network	Supplying pharmacy is part of a pharmacy network
Coordinating pharmacist	No	No	No	Yes

FTE = full time equivalent, CP = coordinating physician, TP = treating physician, ND = no data.

**Table 2 antibiotics-15-00005-t002:** Participants’ characteristics.

Stakeholder Category	Number	Gender % Female	Years of NH Involvement Median, Range
Pharmacist	10	80	10, 1–21
TP	3	33	0.5, 0.5–3
CP	6	17	13.5, 2.5–20
Nurse	16	88	9, 1.5–37
Nurse aide	10	90	3.75, 1–20
Management	5	100	17, 14–30
Resident	9	78	1, 0.25–7
Relative	4	100	1.5, 0.25–3

NH = nursing home, TP = treating physician, CP = coordinating physician.

**Table 3 antibiotics-15-00005-t003:** Key behavioural determinants for appropriate prescribing of antibiotics related to UTIs in NHs.

TDF Domain	Key Behavioural Determinant	Stakeholder	Illustrative Quote
Knowledge	Lack of knowledge on guidelines	NT	“I don’t think we know the scientific guidelines; I don’t anyway (laughs).” (NH2N2) “Indeed, we don’t have any insight into that.” (NH2NA3)
Memory, attention and decision process	Lack of self-reflection and monitoring	NT, P, TP, CP	“I think it [the antibiotic] does get administered too quickly. That it is given too much. That maybe as physicians, we act too quickly. I do think quite frequently.” (NH3TP)
Emotion	Fear of missing complications	NT, TP, CP, R	“I’m more afraid of missing something because these patients [NH residents] are more prone to complications and serious illness. They have little reserve, so if things go wrong, they often recover poorly, with a lasting decline in quality of life.” (CP1)
	Feelings of powerlessness	NT, P, R, M	“Maybe I feel a bit powerless somehow because we cannot directly change it [antibiotic prescribing] ourselves.” (NH2P)
Goals	Prioritising residents’ comfort	NT, R, M	“The main thing is that she [the NH resident] feels well again soon.” (NH3Rel1)
Social influence	Hierarchical relations with TP being dominant	NT, P, TP, CP, R	“And if it comes from the TP, then we just do it.” (NH3N1) “You just follow it. We can’t say no anyway.” (NH3NA1)
	Social pressure to prescribe	NT, R	“So we do have some family members here…‘oh there is an UTI; let the doctor... just get something started.’ Yes, they can be very compelling, you know. And they can also pressure doctors very forcefully.” (NH4N1)
Environmental context and resources	Nursing home as a challenging context	NT, P, TP, CP, M	“Yes, at the moment, as pharmacists, we generally receive far too little information. In fact, we don’t even get the indication for why something is prescribed—that’s where it already starts.” (P3)

TDF = Theoretical Domains Framework, NT = nursing team, N = nurse, NA = nurse aide, P = pharmacist, TP = treating physician, CP = coordinating physician, M = nursing home management, R = residents and relatives, Rel = relative.

## Data Availability

The datasets generated and/or analysed during the current study are not publicly available to preserve the anonymity of the participants but are available from the corresponding author on reasonable request.

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
