# Peer review of "Behavioural Determinants of Appropriate Antibiotic Prescribing for Urinary Tract Infections in Nursing Homes: A Qualitative Study of Stakeholders’ Perspectives"

_antibiotics, 2025, doi:10.3390/antibiotics15010005_

Round 1

Reviewer 1 Report

Comments and Suggestions for Authors

Thank you for the opportunity to review the manuscript " Behavioural determinants of appropriate antibiotic prescribing for urinary tract infections in nursing homes: a qualitative study of stakeholders’ perspective “.

The article discusses the important issue of antibiotic prescriptions by healthcare workers for elderly individuals residing in nursing homes.

There are some important issues that the authors need to address before the manuscript can be considered for publication.

Introduction

In the introduction, in the first paragraph, the sentences do not flow smoothly and coherently.

Line 45- Different (inter)national authorities, such as the World Health Organization, have therefore called for urgent action. What Action?

The introduction does not provide epidemiological data on antibiotic prescribing in nursing homes across Europe; instead, it focuses specifically on Belgium. This limitation makes the article more suitable for a national journal rather than an international one.

Additionally, the article should include the behavioral factors that influence the appropriate prescription of antibiotics for urinary tract infections. It is also important to address the significance of teamwork within nursing homes for the elderly. Furthermore, data specific to Belgium should be consolidated into a single paragraph for clarity.

This adjustment would enhance the article's relevance for a broader, international audience instead of a national one.

The remaining sections of the manuscript are described in detail.

Author Response

C1: Thank you for the opportunity to review the manuscript " Behavioural determinants of appropriate 
antibiotic prescribing for urinary tract infections in nursing homes: a qualitative study of 
stakeholders’ perspective “. The article discusses the important issue of antibiotic prescriptions 
by healthcare workers for elderly individuals residing in nursing homes. There are some important 
issues that the authors need to address before the manuscript can be considered for publication.

A1: Dear reviewer, thank you very much for reading our manuscript and for your valuable comments. We have 
addressed each of them point by point.

C2: Introduction
In the introduction, in the first paragraph, the sentences do not flow smoothly and coherently. Line 
45- Different (inter)national authorities, such as the World Health Organization, have therefore 
called for urgent action. What Action?

A2: Thank you for this helpful comment. We have revised the first paragraph of the Introduction (line 
40-49) to improve the coherence and overall readability of the text. In response to the question 
regarding the “urgent action”, we clarified this by explicitly referring to antimicrobial stewardship. 
Antimicrobial stewardship is one of the key measures highlighted in both national and 
international action plans to address antimicrobial resistance.

C3: The introduction does not provide epidemiological data on antibiotic prescribing in nursing homes 
across Europe; instead, it focuses specifically on Belgium. This limitation makes the article more 
suitable for a national journal rather than an international one.

A3: We added the international report of the HALT-4 study (line 51-57) and incorporated European 
epidemiological data on the prevalence and indications of antimicrobial therapies in nursing 
homes. However, to properly contextualise the relevance of this research within Belgium, we 
consider it important to also refer to national studies that have previously been conducted.

C4: Additionally, the article should include the behavioral factors that influence the appropriate 
prescription of antibiotics for urinary tract infections. 

A4: We do not fully understand this comment, as identifying the behavioural determinants that 
influence appropriate antibiotic prescribing for urinary tract infections is precisely the aim of our 
study. These factors are explicitly analysed and presented in the Results and further interpreted 
in the Discussion.

C5: It is also important to address the significance of teamwork within nursing homes for the elderly.

A5: Indeed, teamwork plays a crucial role within nursing homes. We addressed the social dynamics 
between the various stakeholders in the Results section at the meso-level, highlighting how 
collaborative interactions shape care processes and antimicrobial stewardship practices.

C6: Furthermore, data specific to Belgium should be consolidated into a single paragraph for clarity.
This adjustment would enhance the article's relevance for a broader, international audience 
instead of a national one.

A6: Macro-contextual factors specific to the Belgian population are difficult to disentangle from an 
insider perspective, making it challenging to determine which elements may not be applicable to 
other countries. This limitation is also acknowledged in the Discussion (line 418-420), where we 
note that some findings may be context-specific: while many insights are transferable, results 
related to particular professional roles should be interpreted with caution outside the Flemish 
context.

C7: The remaining sections of the manuscript are described in detail. 

Reviewer 2 Report

Comments and Suggestions for Authors

This is a well written and thoughtful study. I have only one question and a few suggestions to improve clarity.

Were the interviews conducted in English? If translations were required, please describe this in your methods and how you've ensured that meaning during translation was maintained. 

Please ensure that your acronyms are clear in the text in the order that they appear. They should be introduced at their first appearance. If the format of this journal is to place the Methods section at the end, the acronym HCP (line 118) needs to be introduced at its first appearance. Likewise TP on line 106 is used before it is introduced.

In the abstract you write about key determinants of "appropriate" antibiotic prescribing (lines 25-30), which is the positive behaviour, but then use the negative behaviour descriptions. e.g. "lack of knowledge on guidelines". I would assume that the lack of knowledge on guidelines would lead to inappropriate antibiotic prescribing, not appropriate. Please check that you are using the correct supporting behaviours throughout the paper and not the opposites which would be the determinants of inappropriate prescribing.

In line 227 when you direct readers to "see below" it would be helpful to add the section numbers after this direction to assist the reader.

A few more illustrative quotes from participants in sections 2.4.1, 2.4.2 and 2.4.3 could improve the clarity of your evidence, but I understand that publishing word limits can make presenting qualitative data a challenge.

There are spelling errors in Additional File 3 on page 3.

Author Response

C1: This is a well written and thoughtful study. I have only one question and a few suggestions to 
improve clarity.

A1: Dear reviewer
We sincerely thank you for taking the time to review our manuscript, for your positive evaluation, 
and for your insightful comments. We have addressed each point in detail.

C2: Were the interviews conducted in English? If translations were required, please describe this in 
your methods and how you've ensured that meaning during translation was maintained.

A2: The interviews were conducted in Dutch, and the quoted excerpts were translated into English for 
reporting purposes. To ensure that the original meaning was accurately preserved, the 
translations were reviewed and validated by a native English speaker who is also proficient in 
Dutch. This translation procedure has been added to the Methods (line 476-479) section and is 
also acknowledged in the revised manuscript (line 548-549).

C3: Please ensure that your acronyms are clear in the text in the order that they appear. They should 
be introduced at their first appearance. If the format of this journal is to place the Methods section 
at the end, the acronym HCP (line 118) needs to be introduced at its first appearance. Likewise TP 
on line 106 is used before it is introduced.

A3: Indeed, the acronyms were originally introduced in the Methods section, which appears at the 
end of the manuscript. To improve clarity, the meanings of the acronyms HCP (line 122) and TP 
(line 110) have now been added at their first occurrence in the main text.

C4: In the abstract you write about key determinants of "appropriate" antibiotic prescribing (lines 25-
30), which is the positive behaviour, but then use the negative behaviour descriptions. e.g. "lack 
of knowledge on guidelines". I would assume that the lack of knowledge on guidelines would lead 
to inappropriate antibiotic prescribing, not appropriate. Please check that you are using the 
correct supporting behaviours throughout the paper and not the opposites which would be the 
determinants of inappropriate prescribing.

A4: We designed the study, and subsequently analysed the data, with the aim of identifying 
determinants of the positive/target behaviour, namely appropriate antibiotic prescribing. The 
themes that emerged from the interviews were therefore classified as either facilitators or barriers 
to this target behaviour. For example, “lack of knowledge on guidelines” was identified as a barrier 
to appropriate prescribing. In this sense, negative formulations reflect obstacles to the desired 
behaviour rather than determinants of inappropriate prescribing. We hope that this explanation 
clarifies our approach.

C5: In line 227 when you direct readers to "see below" it would be helpful to add the section numbers 
after this direction to assist the reader.

A5: Thank-you for this comment. We have now added the section numbers.

C6: A few more illustrative quotes from participants in sections 2.4.1, 2.4.2 and 2.4.3 could improve 
the clarity of your evidence, but I understand that publishing word limits can make presenting 
qualitative data a challenge.

A6: Since the journal has no strict word limit, we added several additional quotes at the end of each 
paragraph. These quotes were also validated (see earlier comment).

C7: There are spelling errors in Additional File 3 on page 3.

A7: We corrected the spelling errors in Additional file 3.

Round 2

Reviewer 1 Report

Comments and Suggestions for Authors

The authors revised the manuscript according to the comments from the reviewer.